# APRIL/BLyS deficient rats prevent donor specific antibody (DSA) production and cell proliferation in rodent kidney transplant model

**Natalie M. Bath**[1]*, **Bret M. Verhoven**[1], **Nancy A. Wilson**[1,2], **Weifeng Zeng**[3], **Weixiong Zhong**[4], **Lauren Coons**[1], **Arjang Djamali**[5], **Robert R. Redfield, III**[6]

**1** Department of Surgery, University of Wisconsin-Madison, Madison, Wisconsin, United States of America, **2** Division of Nephrology, Department of Medicine, University of Wisconsin-Madison, Madison, Wisconsin, United States of America, **3** Division of Plastic Surgery, Department of Surgery, University of Wisconsin-Madison, Madison, Wisconsin, United States of America, **4** Department of Pathology, University of Wisconsin-Madison, Madison, Wisconsin, United States of America, **5** Department of Medicine, Maine Medical Center, Portland, Maine, United States of America, **6** Division of Transplant, Department of Surgery, Hospital of the University of Pennsylvania, Philadelphia, Pennsylvania, United States of America

* nataliebath124@gmail.com

**Data Availability Statement:** All relevant data are within the paper and its Supporting Information files.

## Abstract

APRIL (A proliferation inducing ligand) and BLyS (B Lymphocyte Stimulator) are two critical survival factors for B lymphocytes and plasma cells, the main source of alloantibody. We sought to characterize the specific effects of these cytokines in a kidney transplant model of antibody mediated rejection (AMR). We engineered APRIL$^{-/-}$ and BLyS$^{-/-}$ Lewis rats using CRISPR/Cas9. APRIL$^{-/-}$ and BLyS$^{-/-}$ rats were sensitized with Brown Norway (BN) blood (complete MHC mismatch). Twenty-one days following sensitization, animals were harvested and collected tissues were analyzed using flow cytometry, ELISPOT, and immunohistochemistry. Flow cross match and a 3 day mixed lymphocyte reaction (MLR) was performed to assess donor specific antibody (DSA) production and T-cell proliferation, respectively. Sensitized dual knock out Lewis rats (APRIL$^{-/-}$/BLyS$^{-/-}$) underwent kidney transplantation and were sacrificed on day 7 post-transplant. Sensitized BLyS$^{-/-}$ had significant decreases in DSA and cell proliferation compared to WT and APRIL$^{-/-}$ (p<0.02). Additionally, BLyS$^{-/-}$ rats had a significant reduction in IgG secreting cells in splenic marginal zone B lymphocytes, and in cell proliferation when challenged with alloantigen compared to WT and APRIL$^{-/-}$. Transplanted APRIL$^{-/-}$/BLyS$^{-/-}$ rodents had significantly less DSA and antibody secreting cells compared to WT (p<0.05); however, this did not translate into a significant difference in AMR seen between groups. In summary, our studies suggest that APRIL and BLyS play a greater role in DSA generation rather than AMR, highlighting the role of cellular pathways that regulate AMR.

**Funding:** RR- KL2 career development award (4KL2TR000428-10), https://ictr.wisc.edu/career-development-awards-2/; American Society of Transplant Surgeons Faculty Development Grant (MSN183242), https://asts.org/asts-foundation/grants-and-eligibility; American College of Surgeons Franklin Martin, MD, FACS Faculty Research Fellowship (MSN192116), https://www.facs.org/member-services/scholarships/research/acsfaculty. The funders had no role in study design, data collection and analysis, decision to publish, or preparation of the manuscript.

**Competing interests:** The authors have declared that no competing interests exist.

## Introduction

Kidney transplantation remains the standard of care in treatment for end stage renal disease. Over the last decade, one-year kidney allograft survival continues to improve and remains above 91%. However, despite the advances made in ten year allograft survival over the last several decades, there still remains much room for improvement [1]. Alloantibody production is a substantial problem due to sensitization (through blood transfusion, pregnancy, or previous transplant) before transplant, which can significantly delay or prevent a patient's transplant [2]. Alloantibody represents a threat post-transplant through antibody mediated rejection (AMR) and donor specific antibody (DSA), the primary causes of long-term graft failure [3]. Due to the fact that thousands of patients rely upon a functioning allograft for survival, there is a critical, unmet need to improve long-term rates of rejection and failure.

B lymphocytes play a complex role in antibody mediated rejection as both antigen presenting cells (APCs) and the primary producer of alloantibody, especially plasma cells, which are terminally differentiated B lymphocytes [4]. APRIL (a proliferation inducing ligand) and BLyS (B lymphocyte stimulator of the TNF family) represent two survival factors for plasma cells and B lymphocytes that are two potential targets. APRIL binds to BCMA (B-cell maturation antigen) and TACI (Transmembrane activator and calcium modulator and cyclophilin ligand interactor) and plays a critical role in plasma cell survival and immunoglobulin class switching [5–7]. BLyS binds with equal affinity to both TACI and BAFF-R (B cell activation factor from the TNF family) and with weaker affinity to BCMA [8]. Once bound to its receptors, BLyS signals to B lymphocytes to undergo maturation, proliferation and ongoing survival [9, 10]. Pharmacologic therapies currently exist to deplete B lymphocytes; however, a long-term solution to effectively treat AMR will likely need to target B lymphocytes at multiple phases of development, which may be achieved through APRIL and BLyS depletion.

To target these B lymphocyte survival factors, we generated BLyS deficient (BLyS$^{-/-}$) and APRIL deficient (APRIL$^{-/-}$) rats using clustered regularly interspaced short palindromic repeats (CRISPR/Cas9) gene editing technology.

Here we present our initial phenotyping of APRIL and BLyS deficient rodents and explore the efficacy of dual knock outs to prevent AMR in a sensitized rodent kidney transplant model.

## Materials and methods

### Animals

Adult (average 10 weeks) Lewis (Envigo) and adult (average 10 weeks) Brown Norway (BN) (Envigo) were housed in the University of Wisconsin Laboratory Animal Facility at WIMR. APRIL deficient (APRIL$^{-/-}$) BLyS deficient (BLyS$^{-/-}$) Lewis rats were generated using CRISPR/Cas9. APRIL$^{-/-}$ and BLyS$^{-/-}$ Lewis rats were then bred to produce APRIL$^{-/-}$/BLyS$^{-/-}$ (dual APRIL and BLyS deficient). All procedures were performed in accordance with the Animal Care and Use Policies at the University of Wisconsin. Animal health including animal deaths, room temperature, 12-hour light/dark cycles, and cage cleaning among other sanitation duties were performed daily by WIMR housing staff. Food and water were available ad libitum. This research was prospectively approved by School of Medicine and Public Health Institutional Animal Care and Use Committee at the University of Wisconsin. Animals were sacrificed via cardiac puncture. All animal experiments, including transplantation and sacrifice, were performed while animals were anesthetized with inhaled isoflurane. Post-transplantation, animals received buprenorphine 0.01–0.5 mg/kg subcutaneous every 12–24 hours until sacrifice. If animals appeared to be suffering despite subcutaneous hydration and buprenorphine, then the animal was sacrificed to alleviate suffering.

Lewis rats were sensitized with 0.5 mL heparinized BN blood (complete major histocompatibility complex (MHC) mismatch) given intravenously via tail vein. Twenty-one days after sensitization animals were sacrificed and tissues were collected for immediate utilization, were stored in 10% formalin for immunohistochemistry (IHC), were snap frozen in liquid nitrogen and stored at -80˚C or were processed to single cells and cryopreserved in liquid nitrogen. Plasma and PBMC (peripheral blood mononuclear cell) were obtained at sacrifice by cardiac puncture into heparinized tubes. Animal groups include: non-sensitized WT (N = 3), non-sensitized APRIL$^{-/-}$ (N = 7), non-sensitized BLyS$^{-/-}$ (N = 3), sensitized WT (N = 4), sensitized APRIL$^{-/-}$ (N = 8), and sensitized BLyS$^{-/-}$ (N = 9).

## Transplant experimental methodology

WT (N = 6) and APRIL$^{-/-}$/BLyS$^{-/-}$ (N = 7) rats were subsequently incorporated into a rodent kidney transplant model. All rodents in this arm of the study were sensitized for 21 days as described above. All rats received CSA at the time of transplant in order to minimize cellular rejection until sacrifice as previously described [11]. Following sensitization, Lewis rats underwent kidney transplantation from a donor BN and bilateral nephrectomy. The technique used for heterotopic renal transplantation was performed as previously described [12]. Animals that underwent transplantation were monitored daily post-transplant. Animals were sacrificed on day 7 post-transplant and tissues were collected for immediate utilization or were stored as described above.

## Generation of APRIL and BLyS knock out rodents

APRIL (APRIL$^{-/-}$) and BLyS deficient (BLyS$^{-/-}$) Lewis rats were generated using CRISPR/Cas9 at UW-Madison Genome Editing and Engineering as previously described [13]. For the BLyS$^{-/-}$ deficient rats, CRISPR/Cas9 guide RNA made a cut that removed the first two exons from the gene, along with upstream untranslated regions, preventing transcription of the remaining sequences. Likewise, for the APRIL$^{-/-}$ rats, the first exon and upstream untranslated regions were removed. No functional protein is expected to be made from either knockout (Fig 1A and 1B; S1 Raw images). Animals were genotyped as described below. Target sites were selected and gRNAs were synthesized via in vitro transcription, followed by column clean up and ethanol precipitation purification. One-cell Lewis fertilized embryos were micro-injected with a mixture of two gRNA (25ng/ul each) and Cas9 protein (40ng/ul; PNA Bio). Animals were genotyped as described below. Injected embryos were transferred into the oviducts of pseudo-pregnant SD recipients and potential founders born. Weanlings were genotyped by amplifying the region targeted for excision, and amplicons indicative of an excision were Sanger sequenced. Founders were identified then bred back to Lewis mates to establish F1s which were characterized as were the founders.

## PCR

Ear snips of Lewis rats were obtained and mixed with genomic lysis buffer and proteinase K solution overnight at 55˚C as previously described [11]. Once DNA pellets were obtained, deionized water was added and samples were then stored at -20˚C. After thawing, forward and reverse primers were provided from UW-Madison Genome Editing and Engineering for APRIL$^{-/-}$ (#166872; #166873) and BLyS$^{-/-}$ (#167285; #167286). Forward and reverse primers, Q5 reaction buffer 5x (MO493S), dNTPs (U1511), and deionized water were added to extracted DNA. DNA samples were then run on gel electrophoresis in usual fashion. Controls for WT, heterozygote, and homozygote were run with each PCR gel electrophoresis sample (Fig 1C; S1 Raw images).

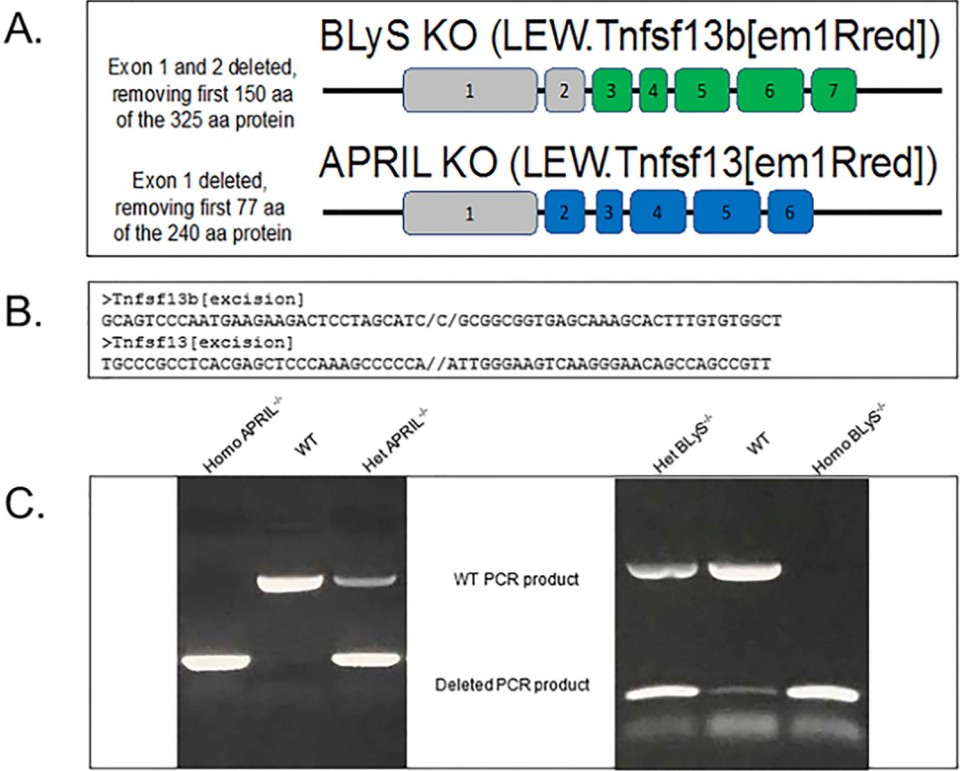

**Fig 1. Generation of APRIL[-/-] and BLyS[-/-] rodents using CRISPR/Cas9.** (A) BLyS and APRIL genes were deleted for early exons, including the transcription and translation start sites. Shaded in grey are exons that were removed using CRISPR/Cas9 guide sequences. These rats are expected to be genetically null for BLyS and APRIL. (B) Genomic sequencing confirmed the excision sites for both BLyS (Tnfsf13b) and APRIL (Tnfsf13) genes. (C) PCR products from heterozygous BLyS and APRIL knockout rats migrates faster than WT PCR product without deletion. Exon 1 and 2 were deleted from the gene for BLyS leaving a smaller PCR product of the expected size for the deletion. Exon 1 was deleted from APRIL. Genomic DNA was sequenced to confirm the excision sites.

## Flow cytometry

Single cell suspensions of splenocytes, bone marrow, PBMC, and mesenteric lymph nodes were prepared from fresh cells. After Ficoll purification, splenocytes, PBMC and bone marrow underwent ACK lysis of red blood cells. After counting and re-suspension in R10 (RPMI with 10% Fetal Calf Serum), 500,000 cells were added to cluster tubes and stained. Cells from each tissue were stained for B lymphocyte subsets, marginal zone (MZ) B lymphocytes, T lymphocytes, and regulatory T cells (Tregs). The antibodies used for the B lymphocyte subsets were as follows: anti-CD38 (clone 14.27 BioLegend), anti-IgD (close MARD3, BioRad), anti-CD45R (B220) (clone HIS24 eBioscience), anti-CD3 (clone 1F4, BD Horizon), anti-CD27 (clone LG.3A10, BD Horizon), anti-CD138 (clone B-A38 abcam), and anti-CD24 (clone ML5, BD Horizon). Antibodies used for the MZ B lymphocytes include anti-MZ B cell (clone HIS57, BD Pharmingen), anti-CD45RA (clone OX-33, BD Pharmingen), and anti-CD3 (clone 1F4, BD Horizon). Antibodies used for the T lymphocyte subsets were as follows: anti-CD8 (clone OX-8, BioLegend), anti-CD45R (B220) (clone HIS24, eBioscience), anti-CD3 (clone 1F4, BD Horizon), and anti-CD4 (clone OX-35, BD Pharmingen). Antibodies used for the Treg stain were anti-CD25 (clone OX-39 BioLegend), anti-CD4 (clone W3/25 BioLegend), anti-CD3 (clone 1F4 BD Horizon), and anti-FoxP3 (clone 150D BioLegend). Flow cytometry was performed on a BD LSR II or BD LSR Fortessa at the UWCCC Flow Cytometry Laboratory and

data analyzed with FlowJo (TreeStar, Inc., Ashland, OR). Ghost Dye Red 780 Viability Dye was used in all stains.

**B lymphocyte subset determination/gating.** Cells were gated to remove non-singlets, then gated through a tight lymphocyte gate based on forward and side scatter. $CD3^+$ cells were removed and were then visualized as IgD *versus* CD45R. From this gate, naïve B lymphocytes ($CD3^-IgD^+CD45R^+CD27^-$) and transitional zone B lymphocytes ($CD3^+IgD^+CD45R^+CD38^+CD24^{++}$) were defined. Memory B lymphocytes were defined as $CD3^-CD27^+CD45R^+$ from the lymphocyte gate. Sample of gating strategy depicting in S1 Fig.

**Marginal zone B lymphocyte gating.** Cells were gated to remove non-singlets, then gated through a tight lymphocyte gate based on forward and side scatter. From the lymphocyte gate, cells were visualized as MZ HIS57 *versus* CD45RA. MZ B lymphocytes were defined as $HIS57^+CD45RA^+$.

**Plasma cell gating.** Cells were gated to remove non-singlets, then through a large gate to ensure capture of the typically larger plasma cells and were subsequently visualized in an IgD *versus* CD45R gate. $IgD^-CD45R^-$ cells were then visualized as CD27 *versus* IgM. Plasma cells were defined as $IgD^-CD45R^-CD27^+IgM^-CD138^+$. Normalized cell counts for plasma cells were calculated based on the large gate, rather than the lymphocyte gate as was used for other lymphocyte populations.

**T lymphocyte subset gating.** Cells were gated to remove non-singlets, then gated through a tight lymphocyte gate based on forward and side scatter. Cells were then visualized as CD4 *versus* CD3 and CD8 *versus* CD3. $CD27^+$ cells were also identified within the $CD3^+CD4^+$ and $CD3^+CD8^+$ populations. T lymphocyte populations were defined as $CD3^+$, $CD3^+CD4^+$, $CD3^+CD8^+$, $CD3^+CD4^+CD27^+$, or $CD3^+CD8^+CD27^+$. Additionally, T follicular helper cells were defined as $CD4^+CCR4^+$, $CD4^+CCR6^+$, $CD4^+CCR10^+$, or $CD4^+CD278^+$.

Macrophages were defined as $CD3^+CD11b^+$ that were forward and side scatter higher and monocytes as $CD3^+CD11b^+$ and in the lymphocyte gate by forward and side scatter.

**Regulatory T cell gating.** Cells were gated to remove non-singlets, then gated through a tight lymphocyte gate based on forward and side scatter. Cells were visualized as CD4 *versus* CD3. $CD4^+CD3^+$ cells were defined as T lymphocytes and then were further gated as CD25 *versus* FOXP3. Regulatory T cells (Tregs) were defined as $CD4^+CD3^+CD25^+FoxP3^+$.

**Flow crossmatch.** Flow crossmatch was performed using donor (Brown Norway) splenocytes freshly isolated from spleen, macerated through a 50 μm sieve, and washed with R10 after red blood cell lysis with ACK. Cells were suspended, counted, and 500,000 cells were aliquoted into cluster tubes for staining. Lewis rat serum from experimental time points was diluted 1:4 in R10 for a total volume of 50 μL, added to BN donor cells for 30 minutes at room temperature, washed, and stained [14, 15]. Antibodies used include: anti-IgG1 (clone RG11/39.4 BD Bioscience), anti-IgG2a (clone RG7/1.30 BD Pharmingen), anti-IgG2c (clone A92-1 BD Pharmingen), anti-CD3 (1F4 BD Horizon), and anti-CD45R (B220) (clone HIS24 eBioscience). Cells were gated to remove non-singlets, through a lymphocyte gate and then a $CD3^+$ or $CD45R^+$ gate was used in order to perform T lymphocyte or B lymphocyte flow crossmatch, respectively. Mean fluorescence intensity (MFI) was determined for the population of interest.

## Histology

Spleen tissue was preserved in 10% formalin for at least 24 hours, processed, paraffin embedded, and cut into 5 μm sections. After deparaffinizing and rehydrating, sections were stained with anti-PAX5 (Abcam, ab140341 polyclonal) antibody overnight. The ImmPRESS HRP reagent and ImmPACT DAB substrate were used to detect anti-PAX5 primary antibody as a

brown pigment. All hematoxylin-eosin C4d slides were reviewed by a transplant pathologist and scored for peritubular capillaritis (ptc), glomerulitis (g), tubulitis (t), vasculitis (v), interstitial inflammation (i), mm (mesangial matrix expansion), ti (total inflammation), i-IFTA (inflammation in area of IFTA), cg (glomerular basement membrane double contours), ct (tubular atrophy), ci (interstitial fibrosis), ah (arteriolar hyalinosis), cv (vascular fibrous intimal thickening), and C4d [15]. AMR and ACR scores were calculated according to Banff 2018 guidelines [16].

## ELISPOT

Single cell suspensions of splenocytes, bone marrow, and mesenteric lymph nodes were prepared from fresh tissue with Ficoll purification. Cells were counted and added to the plate (3654-WP-10 ManTech) previously coated with 10 μg/mL anti-IgG (315-005-046, JacksonImmuno Research Laboratories) or 10 μg/mL anti-IgM (315-005-049, JacksonImmuno Research Laboratories) in bicarbonate coating buffer, washed, and then blocked. Cells were incubated overnight at 37°C in a 5% $CO_2$ incubator. The following day cells were removed and the plate was washed 5 times with PBS. Then 0.1 μg/mL biotinylated anti-IgG (315-065-046, JacksonImmuno Research Laboratories) or 0.1 μg/mL biotinylated anti-IgM (315-065-049, JacksonImmuno Research Laboratories) in PBS/0.1% Tween20 was added at 100 μL per well and incubated for 2 hours at RT. After washing, ExtrAvidin-ALP (1:1000) (Sigma Aldrich, E2636) in PBS was added at 100 μL per well, incubated 1 hour at RT. After washing, 100 μL per well of substrate (3650–10 BCIP/NBT-plus MabTech) was added and allowed to develop until distinct spots emerged. Color development was stopped by washing extensively in tap water. After drying, spots were counted by hand using a dissecting microscope.

## Mixed lymphocyte reaction (MLR)

Lewis WT, APRIL$^{-/-}$, BLyS$^{-/-}$ splenocytes were purified as described above. BN spleen tissue was irradiated with 2000 rad to prevent proliferation. 1 x $10^5$ of Lewis WT, APRIL$^{-/-}$, and BLyS$^{-/-}$ splenocytes were mixed individually with 1 x $10^5$ of irradiated BN splenocytes. Cells were mixed in a 6 well plate and incubated for 72 hours. At harvest, cells were tested for proliferation using a CCK-8 proliferation assay (Dojindo Laboratories). Reported data represents results obtained from serial dilutions.

## Statistics

Statistics were performed using the statistical packages that are part of Prism 7 for Windows, v 7.0b. ANOVA and student's T-test were primarily used. *P* values of 0.05 or less were considered significant. Statistical calculations to determine power were determined prior to implementation of this experiment.

## Results

### BLyS$^{-/-}$ rodents significantly decreased IgM and IgG secreting cells

Antibody secreting cells are the primary source of alloantibody in kidney transplantation; therefore, we wanted to determine the effect of APRIL$^{-/-}$ and BLyS$^{-/-}$ on antibody producing capabilities. IgM and IgG secreting cells were assessed in non-sensitized animals using ELISPOT (Fig 2). BLyS$^{-/-}$ rodents significantly depleted both IgM and IgG secreting cells in all tissues compared to both WT and APRIL$^{-/-}$ (p<0.05). APRIL$^{-/-}$ only significantly decreased IgM secreting cells compared to WT in lymph node (p<0.008). Interestingly, spleen IgM secreting cells were significantly elevated in APRIL$^{-/-}$ compared to WT (p<0.02).

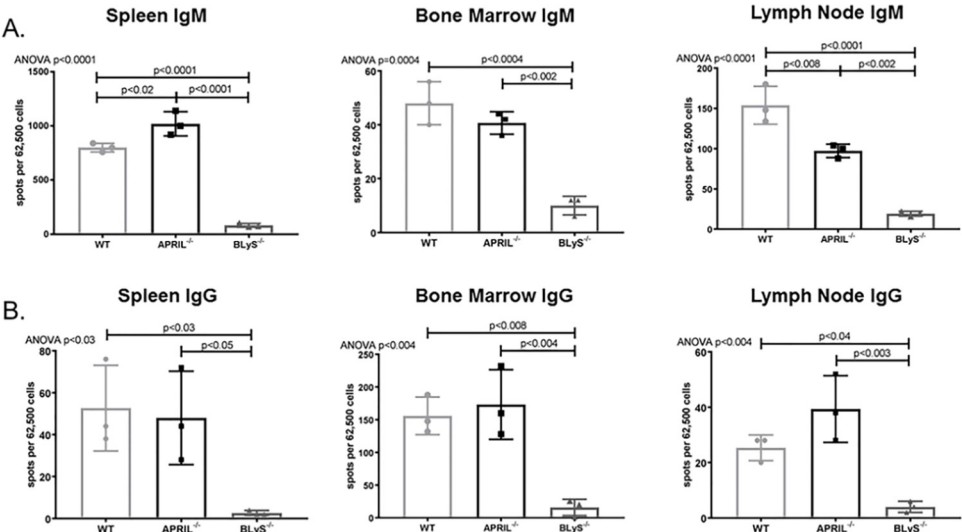

**Fig 2. BLyS⁻/⁻ significantly decreased IgM and IgG secreting cells.** (A) IgM and (B) IgG secreting cells were measured in spleen, bone marrow, and lymph node tissues using ELISPOT. Each graph shows number of spots per 62,500 lymphocytes. Spots represent antibody secreting cells. Data analyzed using ANOVA. BLyS⁻/⁻ significantly depleted IgM and IgG secreting cells in all tissues compared to both WT and APRIL⁻/⁻. This effect was not seen in APRIL⁻/⁻ animals.

## Depleted antibody secreting cells did not result in decreased donor specific antibody production in non-sensitized animals

To investigate whether decreased levels of IgM and IgG secreting cells correlated with a decrease in donor specific antibody (DSA) production, we performed a B (CD45R⁺) and T (CD3⁺) flow crossmatch in non-sensitized animals. Despite the difference noted in antibody secreting cells between groups, DSA production overall was not significantly changed in APRIL⁻/⁻ and BLyS⁻/⁻ compared to WT. CD3⁺IgG1⁺ in BLyS⁻/⁻ was the only DSA significantly decreased compared to APRIL⁻/⁻ (p<0.03) (Fig 3). This finding suggests that although antibody secreting cells in BLyS⁻/⁻ were profoundly decreased compared to WT and APRIL⁻/⁻, antibody production is not significantly altered in a non-sensitized model compared to WT.

## APRIL⁻/⁻ and BLyS⁻/⁻ lymphocyte proliferation significantly decreased when challenged with alloantigen

We performed a mixed lymphocyte reaction (MLR) using irradiated BN lymphocytes (stimulator) and non-sensitized BN (syngeneic), WT (baseline control), APRIL⁻/⁻, and BLyS⁻/⁻ lymphocytes (responder) isolated from spleen. WT, APRIL⁻/⁻, and BLyS⁻/⁻ lymphocytes were all from Lewis rodents, as previously mentioned. Proliferation was then measured after three days to assess the functionality of lymphocytes present. When stimulated with alloantigen, BLyS⁻/⁻ lymphocyte proliferation was significantly decreased compared to both WT and APRIL⁻/⁻ (p<0.005) (Fig 4). Additionally, APRIL⁻/⁻ demonstrated significantly less lymphocyte proliferation compared to WT (p<0.01). These data suggest that when in an environment devoid of B lymphocyte survival factors, lymphocytes become anergic when challenged with alloantigen.

## B lymphocytes in splenic germinal centers significantly depleted in BLyS⁻/⁻

B lymphocyte populations in splenic germinal centers were characterized using flow cytometry and IHC with anti-PAX5 antibody (Fig 5). Non-sensitized BLyS⁻/⁻ (Fig 5C) demonstrated

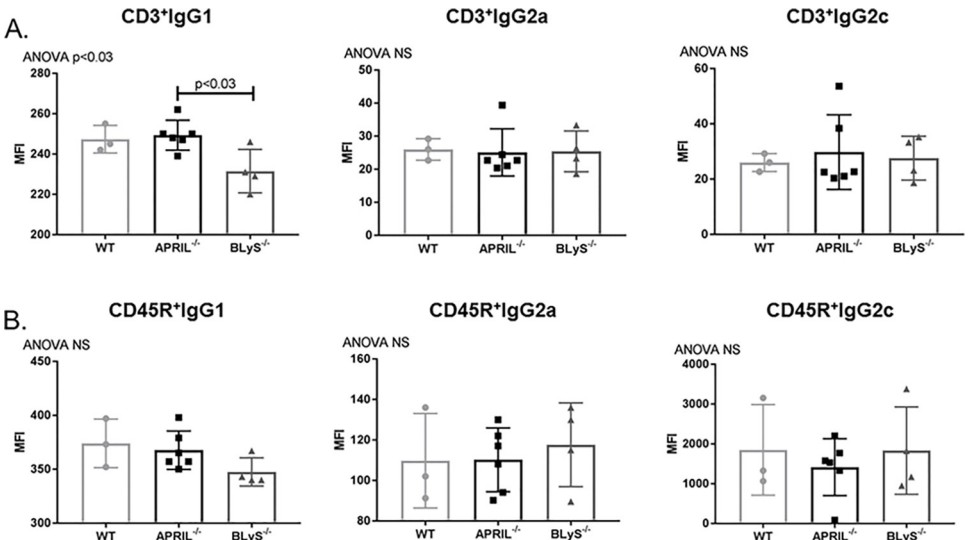

**Fig 3. Depleted antibody secreting cells did not result in decreased donor specific antibody in non-sensitized animals.** Mean fluorescence intensity (MFI) was determined for the population of interest. Data analyzed using ANOVA. (A) T (CD3$^+$) and (B) B (CD45R$^+$) cell flow crossmatch was performed in non-sensitized animals. IgG1$^+$ in the T cell flow crossmatch was the only DSA that BLyS$^{-/-}$ significantly decreased compared to APRIL$^{-/-}$ (p<0.03). No other difference was noted.

significantly disrupted germinal centers compared to both WT and APRIL$^{-/-}$ (p<0.0001) (Fig 5D). Normal architecture was noted in non-sensitized WT and APRIL$^{-/-}$ (Fig 5A and 5B). Flow cytometry was used to identify which B lymphocytes in splenic germinal centers were in fact marginal zone (MZ) B lymphocytes. MZ B lymphocytes (CD45RA$^+$HIS57$^+$) were decreased compared to WT although this was not significant (p = 0.058) (Fig 5E).

## Sensitized BLyS$^{-/-}$ significantly reduces IgG secreting cells

As a result of the significant reduction in antibody secreting cells and splenic germinal center disruption in BLyS$^{-/-}$ animals, we investigated how B lymphocyte populations and functionality were altered in a sensitized model. To perform this task, WT, APRIL$^{-/-}$, and BLyS$^{-/-}$ animals were sensitized via tail vein injection with BN blood (complete MHC mismatch). We allowed three weeks to pass in order to allow sufficient time for alloantibody to form. When evaluating antibody secreting cells, BLyS$^{-/-}$ animals had significantly depleted IgG secreting cells in all tissues compared to both sensitized WT and APRIL$^{-/-}$ animals (p<0.04) (Fig 6B). Spleen IgM secreting cells were also significantly decreased in BLyS$^{-/-}$ compared to WT and APRIL$^{-/-}$ (p<0.02). APRIL$^{-/-}$ significantly decreased spleen IgG secreting cells compared to WT (p<0.05).

## Sensitized BLyS$^{-/-}$ produces significantly less alloantibody against foreign antigens

Although not seen in the non-sensitized model, we wanted to determine if the reduction in antibody secreting cells would correlate with a decrease in DSA when animals were previously exposed to alloantigen (i.e., sensitized). We performed T (CD3$^+$) and B (CD45R$^+$) flow crossmatch as previously described using BN splenocytes and serum from sensitized WT, APRIL$^{-/-}$, and BLyS$^{-/-}$. Interestingly, in the sensitized model, APRIL$^{-/-}$ and BLyS$^{-/-}$ animals profoundly reduced IgG2a and IgG2c in the T cell flow crossmatch (p<0.02) and IgG2a in the B cell flow crossmatch (p = 0.04) (Fig 7). BLyS$^{-/-}$ demonstrated the lowest DSA production for all subsets

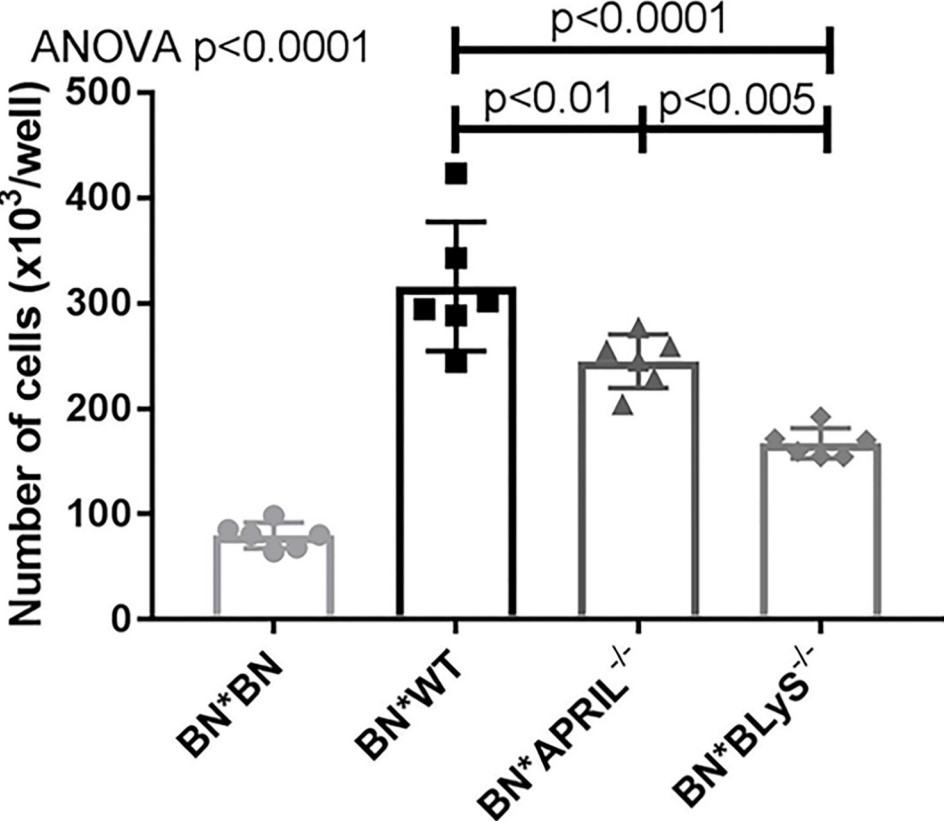

**Fig 4. APRIL$^{-/-}$ and BLyS$^{-/-}$ lymphocyte proliferation significantly decreased when challenged with alloantigen.** Lymphocytes were isolated from BN spleen (stimulator), irradiated, and incubated with lymphocytes from BN rodents, WT Lewis, APRIL$^{-/-}$ Lewis, and BLyS$^{-/-}$ Lewis rodents for 3 days. Data analyzed using ANOVA. BLyS$^{-/-}$ demonstrated significantly less lymphocyte proliferation compared to both WT and APRIL$^{-/-}$ ($p < 0.005$).

and demonstrated a significantly superior prevention of DSA production compared to APRIL$^{-/-}$ for CD3+IgG1, CD45R+IgG1, and CD45R+IgG2c ($p < 0.009$). Therefore, when BLyS is not available to B lymphocytes for utilization, less alloantibody is produced against foreign antigens.

## Sensitized BLyS$^{-/-}$ rodents demonstrated significantly fewer naïve and MZ B lymphocytes compared to sensitized WT and APRIL$^{-/-}$ rodents

Prior to developing into plasmablasts and plasma cells, B lymphocytes reach maturity and reside in secondary lymphoid organs as naïve B lymphocytes [17]. After determining BLyS$^{-/-}$ animals reduced antibody secreting cells present, we wanted to determine if an environment void of B lymphocyte survival factors altered other populations. Sensitized BLyS$^{-/-}$ was found to significantly reduce naïve B lymphocyte populations in all tissues compared to APRIL$^{-/-}$ and in spleen, bone marrow, and lymph nodes compared to WT ($p < 0.04$) (Fig 8A). Similarly, sensitized BLyS$^{-/-}$ demonstrated significant decreases in spleen MZ B lymphocytes compared to sensitized WT ($p < 0.006$), and in all tissues sensitized BLyS$^{-/-}$ had fewer MZ B lymphocytes compared to APRIL$^{-/-}$ ($p < 0.05$) (Fig 8B). Memory B lymphocytes (CD3$^-$CD27$^+$CD45R$^+$), responsible for rapid antibody production in response to alloantigen, were decreased in BLyS$^{-/-}$ throughout all lymphoid tissues compared to WT and APRIL$^{-/-}$; however, these differences were not significant (Fig 8C). Plasma cells (IgD$^-$CD45R$^-$CD27$^+$IgM$^-$CD138$^+$), which rely on APRIL for survival, were decreased in APRIL$^{-/-}$ in PBMCs; however, this finding was not

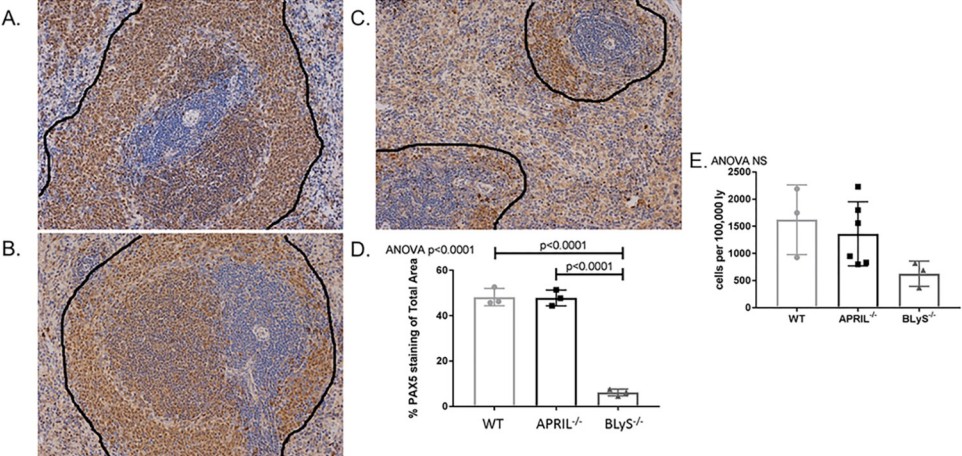

**Fig 5. B lymphocytes in splenic germinal centers significantly depleted in BLyS⁻/⁻.** Anti-PAX5 was used to identify B lymphocytes (brown) surrounding T lymphocytes (blue) in spleen germinal centers in (A) WT, (B) APRIL⁻/⁻, and (C) BLyS⁻/⁻. WT demonstrates normal architecture with the black outlining the germinal center. (D) Graph depicts percentage of total area of spleen staining for anti-PAX5. Significant disruption was noted in BLyS⁻/⁻ compared to WT and APRIL⁻/⁻. (E) Flow cytometry was used to identify which B lymphocytes in splenic germinal centers were marginal zone (MZ) B lymphocytes. MZ B lymphocytes were defined as CD45RA⁺HIS57⁺. Data analyzed using ANOVA.

significant and not seen in other tissues (Fig 8D). These data suggest that although B lymphocyte populations that respond early in immune responses, long-lived immune cells such as memory B lymphocytes were not significantly altered by lack of BLyS or APRIL in the environment.

## Bone marrow, lymph node, and PBMC transitional zone (TZ) B lymphocytes preserved in BLyS⁻/⁻ depleted environment

Once bound to its receptors on immature B lymphocytes, BLyS signals TZ B lymphocytes to mature into naïve B lymphocytes [4]. TZ B lymphocytes have also been found to be tolerogenic

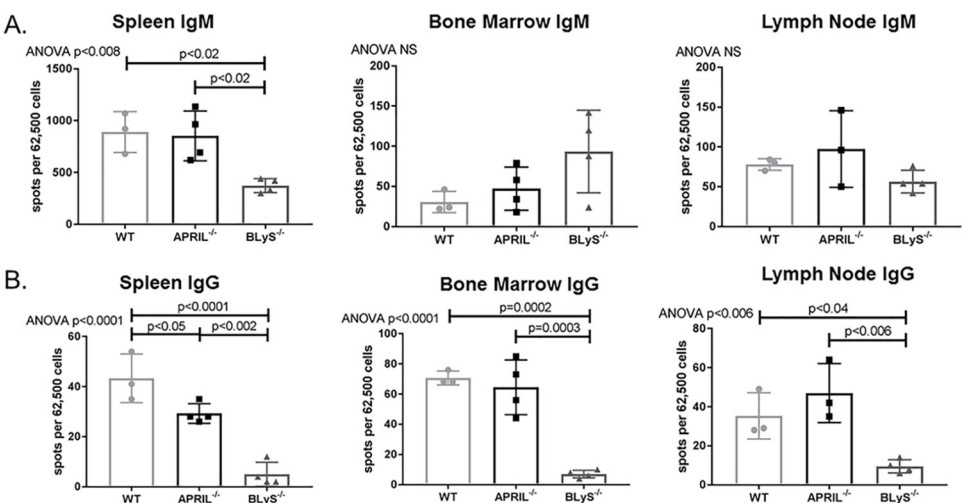

**Fig 6. IgG secreting cells significantly depleted in sensitized BLyS⁻/⁻ animals.** (A) IgM and (B) IgG secreting cells were measured in spleen, bone marrow, and lymph node tissues using ELISPOT. Each graph shows number of spots per 62,500 lymphocytes. Spots represent antibody secreting cells. Data analyzed using ANOVA.

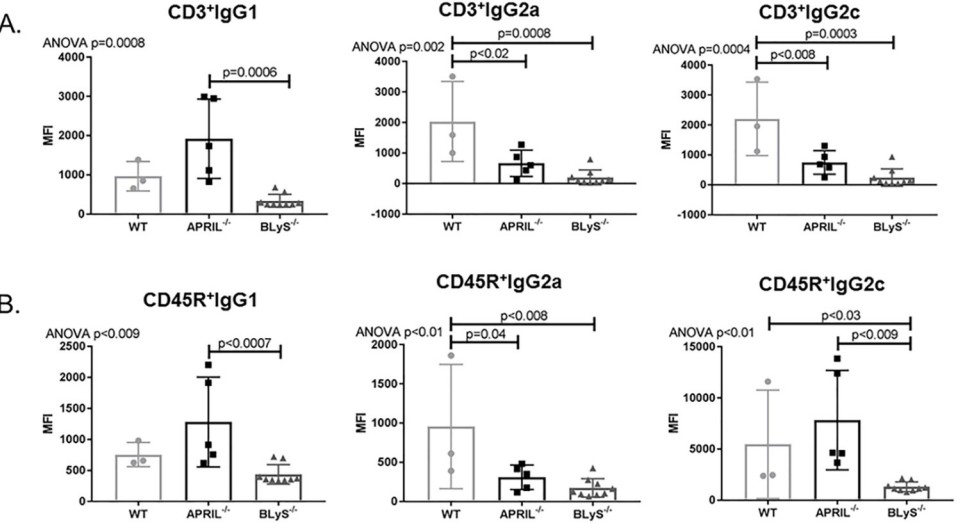

**Fig 7. Sensitized BLyS<sup>-/-</sup> produce significantly less alloantibody against foreign antigens.** (A) T (CD3$^+$) and (B) B (CD45R$^+$) cell flow crossmatch was performed in sensitized animals. Mean fluorescence intensity (MFI) was determined for the population of interest. Data analyzed using ANOVA. Sensitized BLyS$^{-/-}$ produced less DSA compared to WT, which was significant for CD3$^+$IgG2a, CD3$^+$IgG2c, CD45R$^+$IgG2a, and CD45R$^+$IgG2c.

[18]. Therefore, it was important to characterize this population. BLyS$^{-/-}$ significantly reduced splenic TZ B lymphocytes compared to both WT and APRIL$^{-/-}$ (p<0.05); however, this population was largely unchanged from WT in other lymphoid tissues (Fig 9). This finding indicates an accumulation at the TZ B lymphocyte stage when BLyS is not present to signal to cells to undergo further maturation. Preservation of TZ B lymphocytes demonstrated in S2 Fig.

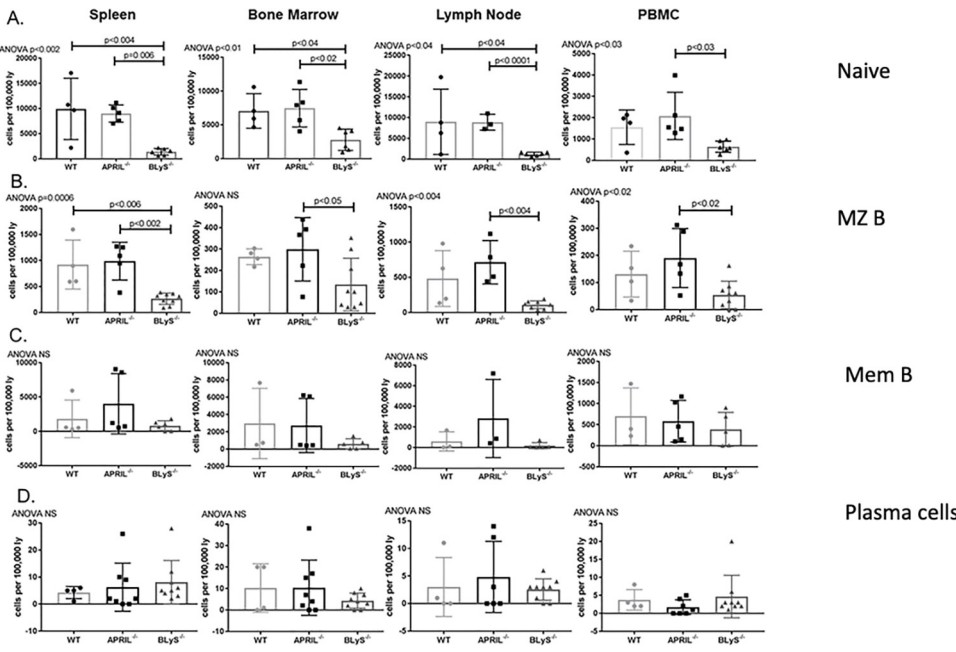

**Fig 8. Sensitized BLyS<sup>-/-</sup> demonstrated significantly fewer naïve and MZ B lymphocytes compared to sensitized WT and APRIL<sup>-/-</sup> rodents.** (A) Naïve B lymphocytes defined as CD3$^-$IgD$^+$CD45R$^+$CD27$^-$. (B) MZ B lymphocytes defined as HIS57$^+$CD45RA$^+$. (C) Memory B lymphocytes defined as CD3$^-$CD27$^+$CD45R$^+$. (D) Plasma cells defined as IgD$^-$CD45R$^-$CD27$^+$IgM$^-$CD138$^+$. Data analyzed using ANOVA.

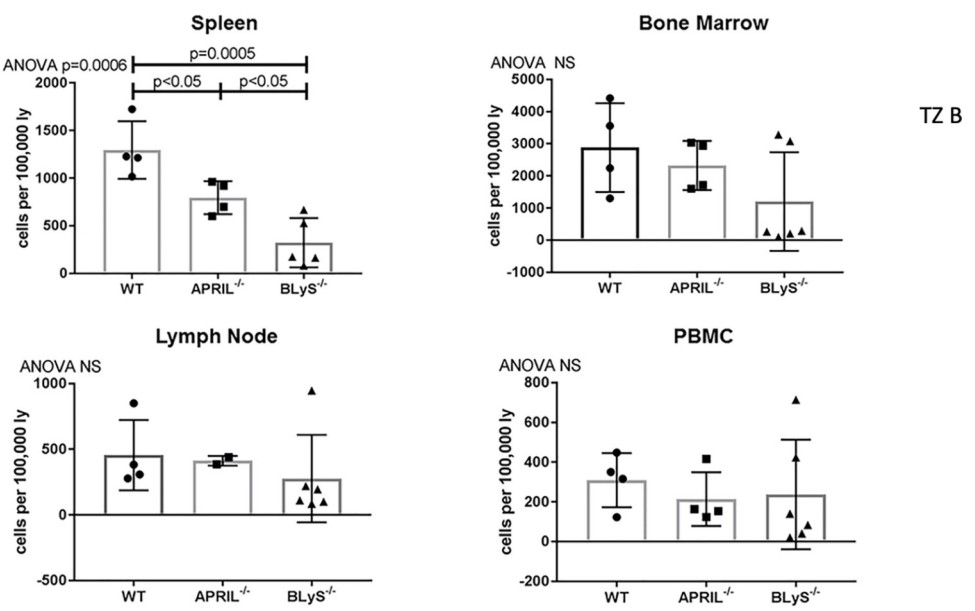

**Fig 9. B lymphocytes accumulate at tolerogenic transitional zone stage when BLyS not present.** Transitional Zone B lymphocytes defined as CD3$^+$IgD$^+$CD45R$^+$CD38$^+$CD24$^{++}$. Data analyzed using ANOVA.

## T lymphocyte populations unchanged by APRIL$^{-/-}$ and BLyS$^{-/-}$

APRIL and BLyS both bind to the TACI-receptor located on activated T lymphocytes, and B lymphocytes also play an important role as an antigen presenting cell to T lymphocytes. Additionally, regulatory T cells (Tregs) have been previously shown to suppress immune responses through tolerance to self-antigens and prevent autoimmune disease [19]. For these reasons, we characterized T lymphocyte populations in these knockout animals. No differences were found between groups in any T lymphocyte subset, which suggests that T lymphocyte populations were not affected by lack of BLyS or APRIL and were also not affected by the smaller populations of B lymphocytes seen in BLyS$^{-/-}$ animals (Fig 10).

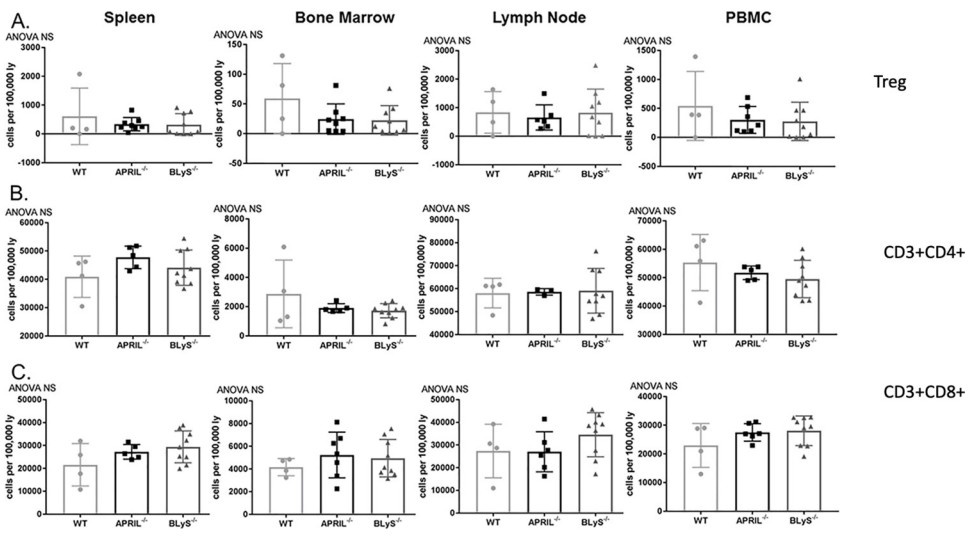

**Fig 10. T lymphocyte populations unchanged by APRIL$^{-/-}$ and BLyS$^{-/-}$.** (A) Regulatory T cells defined as CD4$^+$CD3$^+$CD25$^+$FoxP3$^+$. (B) CD3+CD4+ T lymphocytes. (C) CD3+CD8+ lymphocytes. Data analyzed using ANOVA.

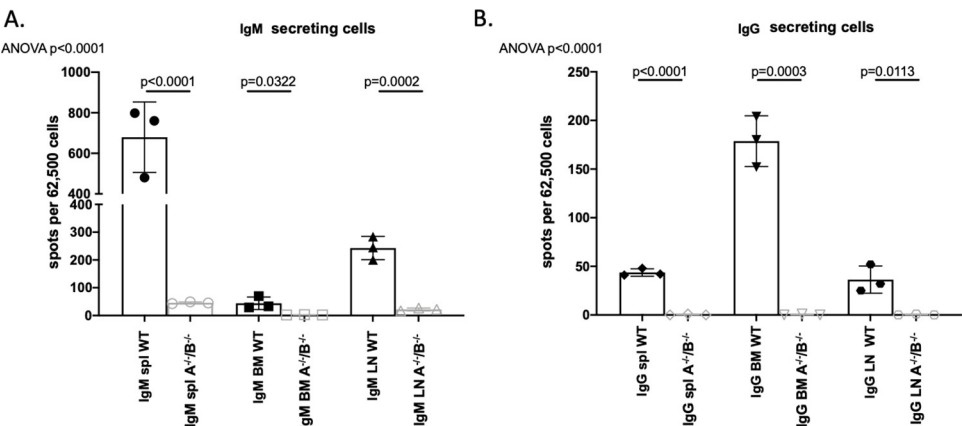

**Fig 11. APRIL<sup>-/-</sup>/BLyS<sup>-/-</sup> rodents found to have significantly fewer antibody secreting cells.** (A) IgM and (B) IgG secreting cells found to be significantly depleted in APRIL<sup>-/-</sup>/BLyS<sup>-/-</sup> transplanted animals across all tissues (p<0.0001). A<sup>-/-</sup>/B<sup>-/-</sup>, APRIL<sup>-/-</sup>/BLyS<sup>-/-</sup>. Data analyzed using ANOVA.

## Sensitized transplanted APRIL<sup>-/-</sup>/BLyS<sup>-/-</sup> rodent model

WT and APRIL<sup>-/-</sup>/BLyS<sup>-/-</sup> Lewis rats underwent kidney transplantation from a donor BN with bilateral nephrectomy. All animals were sensitized for 21 days, received a course of CSA beginning on post-operative day 0, and were sacrificed on post-operative day 7. Tissues were collected for kidney pathology, immunohistochemistry, flow cytometry, and flow crossmatch.

## APRIL<sup>-/-</sup>/BLyS<sup>-/-</sup> rodents found to have significantly fewer antibody secreting cells and less DSA production

As antibody secreting cells depend on survival factors such as APRIL and BLyS to function, we assessed the presence of IgM and IgG secreting cells in transplanted animals via ELISPOT assays. Across all tissues, APRIL<sup>-/-</sup>/BLyS<sup>-/-</sup> rodents produced significantly fewer IgM and IgG secreting cells compared to WT, which was most pronounced in splenic tissue (p<0.0001) (Fig 11).

B (CD45R<sup>+</sup>) and T (CD3<sup>+</sup>) lymphocyte flow crossmatch was performed in order to determine if dual targeting in APRIL and BLyS resulted in decreased DSA production. MFI for IgG1, IgG2a, IgG2b, IgG2c, and IgM were measured in both B and T lymphocyte flow crossmatch. Overall, APRIL<sup>-/-</sup>/BLyS<sup>-/-</sup> rodents produced significantly less DSA in both the B and T lymphocyte flow crossmatch (Fig 12). Specifically, APRIL<sup>-/-</sup>/BLyS<sup>-/-</sup> rodents saw significantly less IgG1 and IgG2a in the T lymphocyte flow crossmatch (p<0.03) and in IgG1 and IgG2a (p<0.03) in the B lymphocyte flow crossmatch. Together, the findings of decreased antibody secreting cells and DSA production support the notion that APRIL and BLyS depletion may decrease antibody production in kidney transplant.

## Complete disruption of splenic architecture demonstrated in APRIL<sup>-/-</sup>/BLyS<sup>-/-</sup> group

In order to further investigate the effect of dual targeting APRIL and BLyS on lymphocyte production, we examined splenic architecture following transplant. Complete disruption of splenic germinal centers was demonstrated in APRIL<sup>-/-</sup>/BLyS<sup>-/-</sup> rodents as evidenced by a significant decrease in B lymphocytes present (p<0.0001). Although not statistically significant, fewer CD3<sup>+</sup> T lymphocytes appeared to be present in APRIL<sup>-/-</sup>/BLyS<sup>-/-</sup> rodents compared to WT (Fig 13).

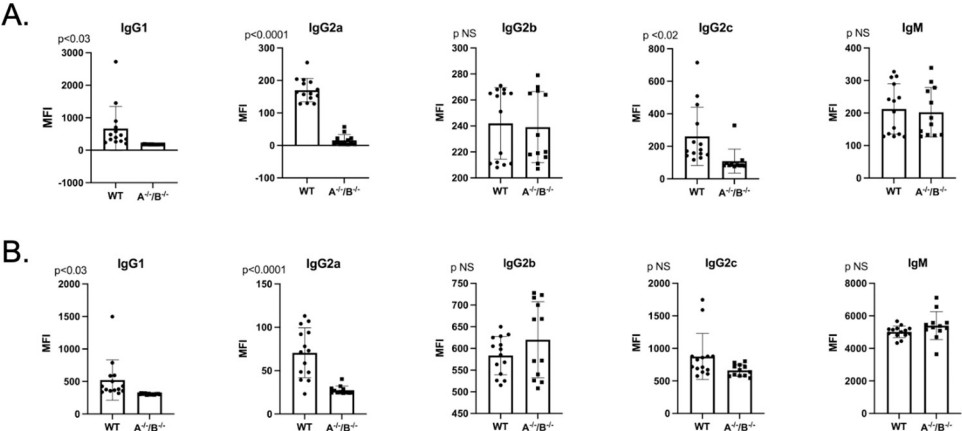

**Fig 12. APRIL$^{-/-}$/BLyS$^{-/-}$ rodents produced significantly less DSA compared to WT.** (A) T (CD3$^+$) and (B) B (CD45$^+$) lymphocyte flow crossmatch was performed as previously described. A$^{-/-}$/B$^{-/-}$, APRIL$^{-/-}$/BLyS$^{-/-}$. Data analyzed using Student's T-test.

## Early-stage and mature B lymphocytes significantly decreased but long-lived B lymphocytes increased in APRIL$^{-/-}$/BLyS$^{-/-}$ group

We assessed B lymphocytes at various stages of development via flow cytometry in order to determine which, if any, cell lines were affected in APRIL$^{-/-}$/BLyS$^{-/-}$ group. B lymphocytes that were assessed included naïve (CD3$^-$IgD$^+$CD45R$^+$CD27$^-$), transitional zone (CD3$^+$IgD$^+$CD45R$^+$CD38$^+$CD24$^{++}$), marginal zone (HIS57$^+$CD45RA$^+$), memory (CD3$^-$CD27$^+$CD45R$^+$), and plasma (IgD$^-$CD45R$^-$CD27$^+$IgM$^-$CD138$^+$) cells. Notably, APRIL$^{-/-}$/BLyS$^{-/-}$ group had significantly fewer splenic, bone marrow, and lymph node naïve B lymphocytes compared to WT (p<0.0001). Depletion of transitional zone B lymphocytes in spleen and lymph node was also seen (p<0.0001). Marginal zone B lymphocytes were significantly depleted in splenic tissue, which supports the finding of splenic marginal zone disruption on histology as previously mentioned. Interestingly, APRIL$^{-/-}$/BLyS$^{-/-}$ group were found

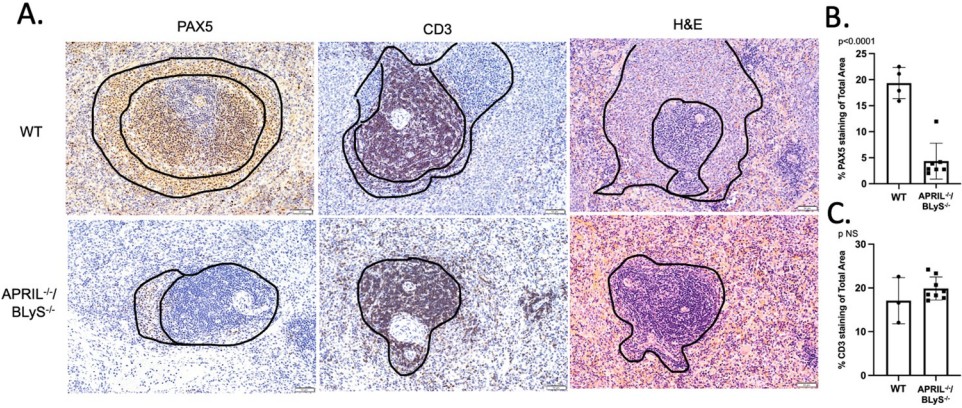

**Fig 13. Splenic germinal centers completely disrupted in APRIL$^{-/-}$/BLyS$^{-/-}$ rodents.** Top row: sensitized transplant control group. Bottom row: sensitized APRIL/BLyS$^{-/-}$ transplant group. (A) Anti-PAX5 was used to identify B lymphocytes (brown) surrounding T lymphocytes (blue) in spleen germinal centers in WT and APRIL$^{-/-}$/BLyS$^{-/-}$. WT demonstrates normal architecture with the black outlining the germinal center. (B) Graph depicts percentage of total area of spleen staining for anti-PAX5 and (C) anti-CD3. Significant disruption was noted in APRIL$^{-/-}$/BLyS$^{-/-}$ compared to WT. Data analyzed with Student's T-test.

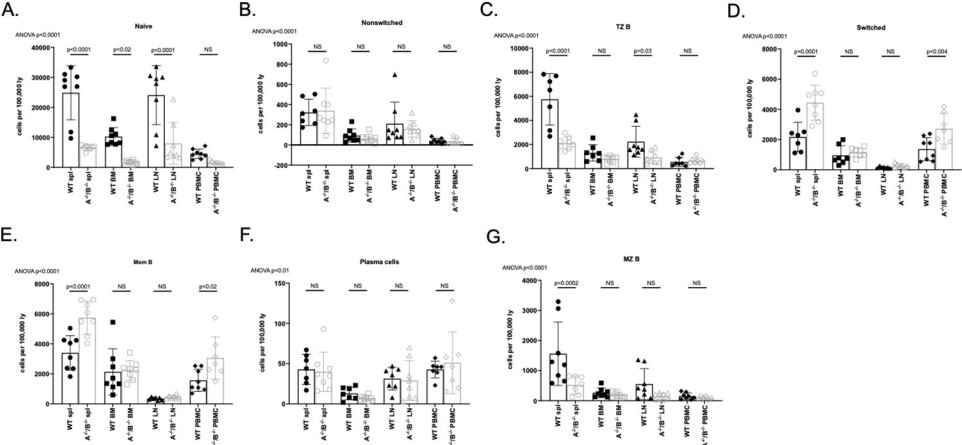

**Fig 14. Early-stage and mature B lymphocytes significantly decreased but long-lived B lymphocytes increased in APRIL<sup>-/-</sup>/BLyS<sup>-/-</sup> group.** Flow cytometry was used to assess B lymphocytes for each transplant group. Each graph shows number of cells per 100,000 lymphocytes. A<sup>-/-</sup>/B<sup>-/-</sup>, APRIL<sup>-/-</sup>/BLyS<sup>-/-</sup>. Data analyzed using ANOVA.

to have a significant increase in long-lived B lymphocytes including memory B and switched B lymphocytes (p<0.01). No difference was seen in plasma cell population between groups (Fig 14).

## T lymphocytes significantly increased in APRIL<sup>-/-</sup>/BLyS<sup>-/-</sup> rodents but no change seen in T follicular helper lymphocytes

Although APRIL<sup>-/-</sup>/BLyS<sup>-/-</sup> group had fewer B lymphocytes, this group was seen to have significantly more T lymphocytes across multiple sub-populations. Both CD3<sup>+</sup>CD4<sup>+</sup> and CD3<sup>+</sup>CD8<sup>+</sup> T lymphocyte populations were significantly increased in APRIL<sup>-/-</sup>/BLyS<sup>-/-</sup> splenic tissue and lymph nodes (p<0.0001) (Fig 15). Similarly, CD4<sup>+</sup>CD27<sup>+</sup> and CD8<sup>+</sup>CD27<sup>+</sup> T lymphocyte populations were significantly increased compared to WT in multiple tissues (p<0.0001). APRIL<sup>-/-</sup>/BLyS<sup>-/-</sup> group also had a larger population of Tregs (CD4<sup>+</sup>CD3<sup>+</sup>CD25<sup>+</sup>FoxP3<sup>+</sup>) compared to WT (p<0.0001). Multiple Tfh cells were assessed as well. While not consistently seen across multiple tissues or populations, APRIL<sup>-/-</sup>/BLyS<sup>-/-</sup>

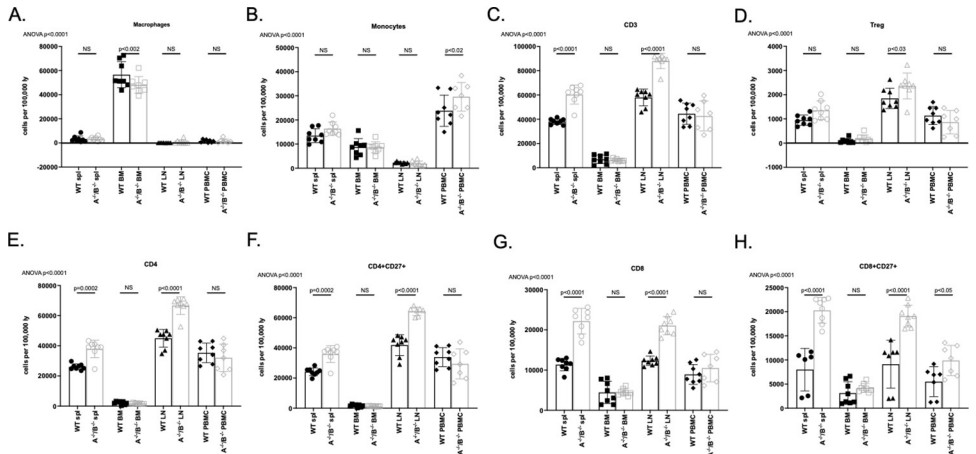

**Fig 15. T lymphocyte populations significantly increased in APRIL<sup>-/-</sup>/BLyS<sup>-/-</sup> group.** Flow cytometry was used to assess T lymphocytes for each transplant group. Each graph shows number of cells per 100,000 lymphocytes. A<sup>-/-</sup>/B<sup>-/-</sup>, APRIL<sup>-/-</sup>/BLyS<sup>-/-</sup>. Data analyzed using ANOVA.

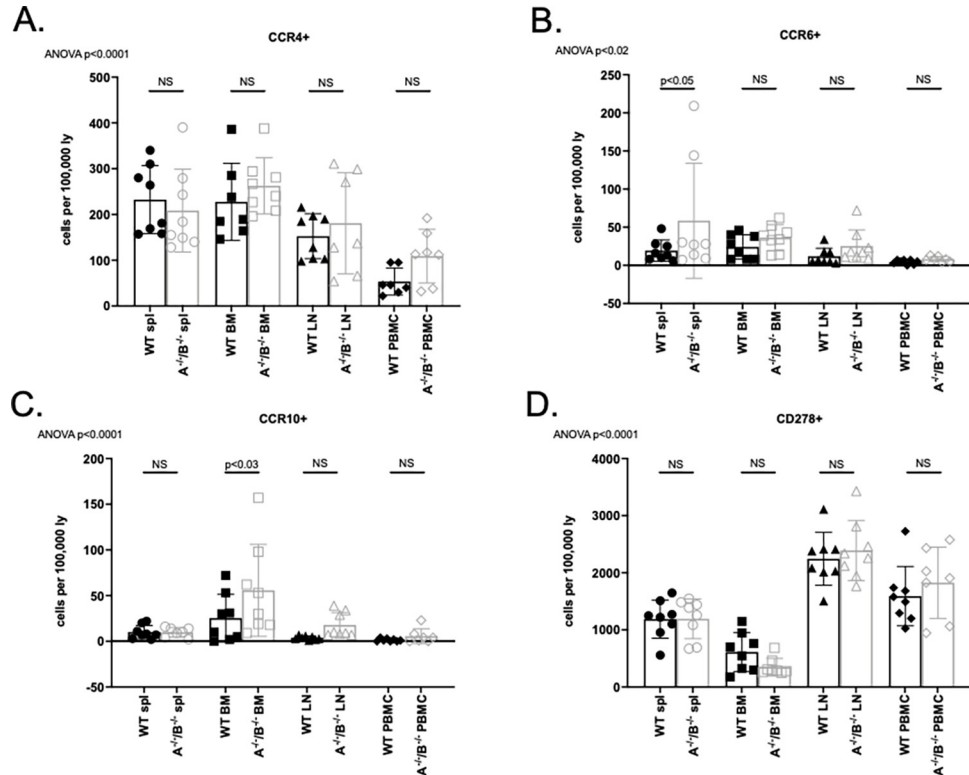

**Fig 16. No change seen in TFH cell populations.** Flow cytometry was used to assess T lymphocytes for each transplant group. Each graph shows number of cells per 100,000 lymphocytes. $A^{-/-}/B^{-/-}$, APRIL$^{-/-}$/BLyS$^{-/-}$. Data analyzed using ANOVA.

group were found to have significantly increased populations of CD4$^+$CCR6$^+$ and CD4$^+$CCR10$^+$ in spleen and bone marrow compared to WT, respectively (Fig 16). Together, these changes suggest that deficiency of APRIL and BLyS may also impact the survival and differentiation of not only B lymphocytes but T lymphocytes as well.

## Less severe ACR and lower incidence AMR seen in APRIL$^{-/-}$/BLyS$^{-/-}$ but no significant difference seen compared to WT

Lastly, we assessed the effect of APRIL and BLyS deficiency on the presence of AMR and ACR in transplanted kidneys. All kidney pathology was reviewed by a transplant pathologist (blinded) and scored for ptc, g, t, v, i, mm, ti, ah, cg, ci, ct, i-IFTA, cv, and C4d. AMR and ACR scores were calculated according to Banff 2018 guidelines [16]. Despite lower scores for most variables seen in APRIL$^{-/-}$/BLyS$^{-/-}$ group, no significant difference was seen in rates of AMR or ACR. One-third (N = 2) of WT animals developed active AMR compared to 14.3% (N = 1) of APRIL$^{-/-}$/BLyS$^{-/-}$ (Table 1). WT group was seen to have more severe grade of ACR with 83.3% (N = 5) having grade I or higher ACR compared to 42.9% (N = 3) APRIL$^{-/-}$/BLyS$^{-/-}$. Similarly, while APRIL$^{-/-}$/BLyS$^{-/-}$ had better kidney function than WT as measured through BUN and creatinine, no statistically significant difference was seen. Representative histologic kidney images are shown in Fig 17.

## Discussion

In this study we demonstrated that targeting APRIL and BLyS in a rodent model results in significant changes in B lymphocyte populations and in a reduction of alloantibody production.

**Table 1.**

| Banff score | WT (N = 6) | | APRIL⁻/⁻/BLyS⁻/⁻ (N = 7) | | |
|---|---|---|---|---|---|
| | Mean | SD | Mean | SD | P |
| t | 2.5 | 1.2 | 1.4 | 1.0 | 0.11 |
| i | 2.3 | 1.2 | 1.4 | 1.0 | 0.16 |
| g | 0.8 | 1.0 | 0.0 | 0.0 | 0.04 |
| ah | 0.0 | 0.0 | 0.0 | 0.0 | - - |
| v | 0.3 | 0.5 | 0.0 | 0.0 | 0.11 |
| ptc | 1.2 | 1.3 | 0.1 | 0.4 | 0.07 |
| ti | 2.3 | 1.2 | 1.4 | 1.0 | 0.16 |
| mi | 0.0 | 0.0 | 0.0 | 0.0 | - - |
| i-IFTA | 0.0 | 0.0 | 0.0 | 0.0 | - - |
| c4d | 0.3 | 0.8 | 0.3 | 0.8 | 0.92 |
| Cg | 0.2 | 0.4 | 0.0 | 0.0 | 0.30 |
| Ct | 0.0 | 0.0 | 0.0 | 0.0 | - - |
| Ci | 0.0 | 0.0 | 0.0 | 0.0 | - - |
| Ah | 0.0 | 0.0 | 0.0 | 0.0 | - - |
| cv | 0.0 | 0.0 | 0.0 | 0.0 | - - |
| mm | 0.0 | 0.0 | 0.0 | 0.0 | - - |
| AMR, % (N) • Active | 33.3 (2) | - - | 14.3 (1) | - - | 0.56 |
| ACR | | | | | 0.17 |
| • Negative | 16.7 (1) | | 14.3 (1) | | |
| • Borderline | 0 (0) | | 42.9 (3) | | |
| • I | 50.0 (3) | | 42.9 (3) | | |
| • II | 33.3(2) | | 0.0 (0) | | |
| • III | 0.0 (0) | | 0.0 (0) | | |
| Kidney function | | | | 6.0 | 0.14 |
| • BUN | 45.7 | 20.0 | 33.0 | 0.1 | 0.45 |
| • Cr | 0.8 | 0.5 | 0.6 | | |

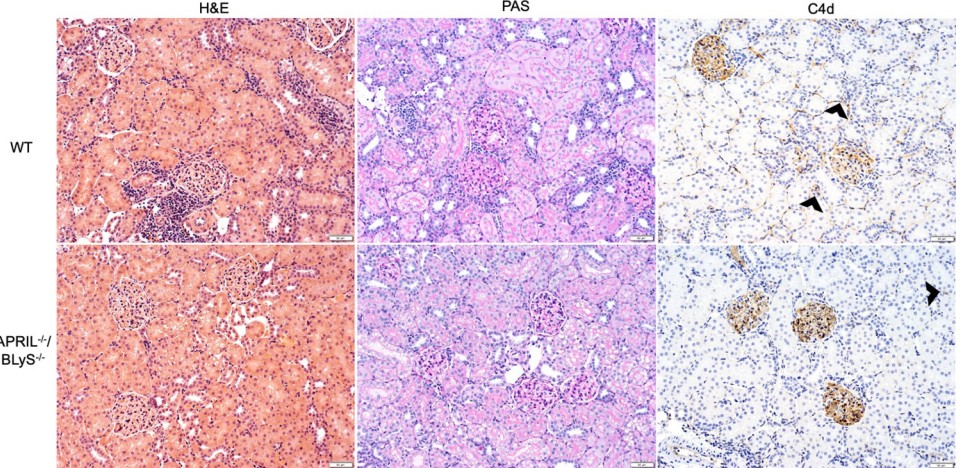

**Fig 17. Kidney pathology.** Top row: WT. Bottom row: APRIL⁻/⁻/BLyS⁻/⁻. From left to right columns: H&E stain, PAS stain, C4d stain. Black arrows indicate C4d deposition.

Changes in B lymphocyte populations resulted in significant decreases in IgM and IgG secreting cells and disruption of splenic germinal centers. Additionally, B lymphocyte function was reduced, which manifested as a reduction of lymphocyte proliferation in BLyS$^{-/-}$ cells when challenged with alloantigen. After characterizing APRIL$^{-/-}$ or BLyS$^{-/-}$ rodents at baseline, we assessed B lymphocyte population and activity changes in a sensitized model. Here, a decrease in IgG secreting cells was seen in BLyS$^{-/-}$ rodents, which then translated into a reduction in DSA production compared to sensitized WT and APRIL$^{-/-}$. APRIL$^{-/-}$ demonstrated less lymphocyte proliferation and marginally less DSA production when challenged with alloantigen. Less dramatic changes were seen in APRIL$^{-/-}$ likely because BLyS was still present in these animals to support B lymphocyte homeostasis. These findings indicate that not only were cell populations altered through the targeting of these survival factors, but the function of existing lymphocyte populations was also significantly reduced.

Importantly, when targeted in a kidney transplant model, APRIL$^{-/-}$/BLyS$^{-/-}$ rodents were found to decrease DSA and had significant changes in both B and T lymphocyte populations. While there was not a significant change in AMR or ACR when compared to WT rodents, animals deficient in APRIL and BLyS saw improved kidney function although not statistically significant. Together, these findings represent important data supporting the use of targeted therapy against B lymphocytes at multiple stages of development.

APRIL and BLyS have long been established as B lymphocyte survival factors, and subsequently have been utilized as therapeutic targets in autoimmune diseases [20–22]. Additionally, increased BLyS levels in systemic lupus erythematosus (SLE) patients and an associated increase in anti-dsDSA suggest the importance of BLyS in the loss of tolerance [23, 24]. Tolerance and appropriate immune system suppression represent a fine balance that must be maintained to achieve long-term graft survival.

Belimumab is a recombinant human monoclonal antibody that specifically binds to soluble BLyS and is an FDA approved treatment for SLE. When combined with standard therapy, belimumab has been shown in multi-center randomized-controlled trials to significantly reduce SLE disease activity and severe flares when compared to placebo with standard therapy [25–27]. More importantly, significantly more patients converted from anti-dsDNA antibody positive to negative when belimumab was given. Interestingly, when antibody titers to previously administered vaccinations were measured, patients who received Belimumab did not have significantly different alloantibody titers. This is consistent with the finding that long-lived memory B cells are not targeted with anti-BLyS therapies [28].

The efficacy of targeting APRIL and BLyS has been established in the autoimmune literature as a method to reduce autoantibody and disease severity. However, the role of APRIL and BLyS as therapeutic targets in transplantation is still to be determined. Belimumab was used as an adjunct to standard-of-care immunosuppression in a phase 2 randomized placebo-controlled trial conducted by Banham et al. Patients who received belimumab were not at increased risk of infection and had a non-significant trend towards a decrease in naïve B cells. Similar to our findings, belimumab-treated patients demonstrated a reduction in de novo IgG formation suggesting a potential use in sensitized patients. Additionally, Agarwal et al. describe the results of a 1-year pilot trial in which belimumab reduced antibodies against selective HLA specificities at various time points; however, anti-BLyS therapy alone did not reduce total anti-HLA allo-antibody levels in highly sensitized patients [29]. This study draws attention to the fact that multi-agent treatments including anti-APRIL or proteasome inhibition may be critical to total DSA desensitization [30].

Clinical trials remain the gold standard to help progress treatment changes in transplantation. Data presented here and in the previously mentioned clinical trials provide further support to the existing data for the role of anti-APRIL and anti-BLyS therapy in transplantation

[31]. Dual therapy has shown promise in the pre-clinical setting and drug safety of current anti-APRIL and anti-BLyS medications have been established independent of each other. Therefore, next steps to move forward combination therapy may involve an open-label pilot study, in which the safety and efficacy of anti-APRIL/BLyS in addition to standard of care immunosuppression is investigated.

## Limitations

The novelty of this study is that we characterized the ability to reduce *allo*antibody in an APRIL and BLyS deficient environment. Although we demonstrated a reduction in alloanti-body with BLyS[-/-], this finding did not correspond with a depletion of memory B or plasma cells. This finding suggests that a multi-targeted approach is likely needed to successfully inhibit long-term humoral responses which ultimately may result in allograft rejection. By targeting both mature B lymphocytes and plasma cells, APRIL and BLyS inhibition could potentially result in long-term changes in B lymphocyte populations. Furthermore, it is possible that by targeting APRIL and BLyS, regulatory B lymphocytes, which play a role in inducing immunological allograft tolerance, are also depleted along with other B lymphocyte subsets. While we did not characterize regulatory B lymphocytes specifically, future studies should investigate changes in this subset as a potential explanation for no overall changes in AMR.

Our data offers further support for the use of APRIL and BLyS inhibition as a method to prevent antibody mediated rejection in transplantation. Within transplant literature, our findings are also supported in a study by Kwun et al. in which APRIL and BLyS were targeted using TACI-Ig (Transmembrane activator and calcium modulator and cyclophilin ligand interactor-Immunoglobulin) in a non-human primate AMR kidney transplant model. TACI-Ig (a recombinant fusion protein that blocks both APRIL and BLyS) use was shown to decrease levels of donor specific antibody post-transplant and decreased pathological findings usually associated with AMR. However, graft survival was only marginally prolonged [20]. These findings suggest that the role of APRIL and BLyS in desensitization and AMR treatment require ongoing investigation.

Lastly, as our ultimate goal was to determine the role of APRIL and BLyS depletion in a sensitized transplant model, our data presented here does not include non-sensitized transplant outcomes in APRIL[-/-]/BLyS[-/-] rodents. Therefore, conclusions presented may only apply to sensitized models.

## Conclusion

Here we have demonstrated that rodents deficient in APRIL and BLyS have decreased IgM and IgG secreting cells, altered B lymphocyte populations, and decreased alloantibody production when stimulated with alloantigen when compared to WT in a preclinical kidney transplant model. Lack of APRIL, the survival factor for terminally differentiated plasma cells, did not result in these changes seen in BLyS[-/-] rodents. However, memory B and plasma cells, which form in late stages of B lymphocyte development, were not depleted in either BLyS[-/-] or APRIL[-/-]. When applied to a sensitized rodent model, targeting BLyS largely impacts B lymphocytes in earlier stages of development, which in turn results in decreased mature B lymphocyte populations and importantly, alloantibody production. In order to successfully target long-lived B lymphocytes such as memory B or plasma cells, combined targeting of both APRIL and BLyS is likely needed.

Our data further support the value of targeting multiple B lymphocyte survival factors in order to successfully reduce anti-HLA antibodies.

## Supporting information

**S1 Raw images. Raw image of PCR gel.**
(TIF)

**S1 Fig. Representative gating strategy for B lymphocytes.** Top row: CD3$^-$ cells are selected from which lymphocytes are defined (forward versus side scatter). From this tight lymphocyte gate, cells are visualized as IgD versus CD45R. Cell populations that arise from here are defined as (1) CD45R$^-$IgD$^-$, (2) CD45R$^+$IgD$^+$, and (3) CD45R$^+$IgD$^-$. Gating in this image all comes from WT rodent. (1) From CD45R$^-$IgD$^-$, CD27$^+$IgM$^-$ cells are gated. These cells are further identified as plasma cells if they are CD138$^+$ or CD38$^+$. Arrows indicate that cell populations originated from population gated in prior graph. (2) From CD45R$^+$IgD$^+$, naïve B lymphocytes are defined as CD27$^-$IgD$^+$ or non-switched B lymphocytes are defined separately from the CD45R$^+$IgD$^+$ as CD27$^+$CD45R$^+$. (3) From CD45R$^+$IgD$^-$, switched B lymphocytes are defined as CD27$^+$IgM$^-$. From CD45R$^+$IgD$^-$, memory B lymphocytes are defined as CD27$^+$CD45R$^+$.
(TIF)

**S2 Fig. Flow cytometry dot plot demonstrating preservation of TZ B lymphocytes in lymph node, bone marrow, and PBMC.** Gating strategy of TZ B lymphocytes demonstrated. After selecting CD3$^-$ lymphocytes from previous gates. Cells are visualized as IgD versus CD45R. IgD$^+$CD45R$^+$ gate (Q2) is selected and visualized as CD38 versus CD24. TZ B lymphocytes are defined as CD38$^+$CD24$^{++}$.
(TIF)

## Author Contributions

**Conceptualization:** Arjang Djamali, Robert R. Redfield, III.

**Data curation:** Natalie M. Bath, Nancy A. Wilson, Weifeng Zeng, Weixiong Zhong, Lauren Coons.

**Formal analysis:** Natalie M. Bath, Bret M. Verhoven, Nancy A. Wilson.

**Funding acquisition:** Robert R. Redfield, III.

**Investigation:** Natalie M. Bath, Arjang Djamali, Robert R. Redfield, III.

**Methodology:** Bret M. Verhoven, Nancy A. Wilson, Arjang Djamali, Robert R. Redfield, III.

**Resources:** Robert R. Redfield, III.

**Supervision:** Arjang Djamali, Robert R. Redfield, III.

**Visualization:** Robert R. Redfield, III.

**Writing – original draft:** Natalie M. Bath.

**Writing – review & editing:** Nancy A. Wilson, Arjang Djamali, Robert R. Redfield, III.

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
