## [Decision Letter · Decision Letter 0]

16 Aug 2022

PONE-D-22-09163

APRIL/BLyS Deficient Rats Prevent Donor Specific Antibody (DSA) Production and Cell Proliferation in Rodent Kidney Transplant Model

PLOS ONE

Dear Dr. Bath,

Thank you for submitting your manuscript to PLOS ONE. After careful consideration, we feel that it has merit but does not fully meet PLOS ONE’s publication criteria as it currently stands. Therefore, we invite you to submit a revised version of the manuscript that addresses the points raised during the review process.

Your manuscript was reviewed by three experts. Please revise it according to their suggestions.

We look forward to receiving your revised manuscript.

Kind regards,

Hodaka Fujii, M.D., Ph.D.

Academic Editor

PLOS ONE

https://journals.plos.org/plosone/s/file?id=ba62/PLOSOne_formatting_sample_title_authors_affiliations.pdf".

“The authors would like to thank the UW CCC Flow Cytometry Shared instrumentation core, including the Shared Instrumentation grant 1S00OD018202-01 Special BD LSR Fortessa, which made possible the purchase and use of the BD LSR Fortessa.”

“RR- KL2 career development award (4KL2TR000428-10), https://ictr.wisc.edu/career-development-awards-2/; American Society of Transplant Surgeons Faculty Development Grant (MSN183242), https://asts.org/asts-foundation/grants-and-eligibility; American College of Surgeons Franklin Martin, MD, FACS Faculty Research Fellowship (MSN192116), https://www.facs.org/member-services/scholarships/research/acsfaculty.

Additional Editor Comments:

In each figure legend, please describe which test (ANOVA or T-test) was used.

Reviewers' comments:

Reviewer's Responses to Questions

**Comments to the Author**

1. Is the manuscript technically sound, and do the data support the conclusions?

Reviewer #1: Yes

Reviewer #2: Yes

Reviewer #3: Yes

2. Has the statistical analysis been performed appropriately and rigorously? 

Reviewer #1: Yes

Reviewer #2: Yes

Reviewer #3: Yes

3. Have the authors made all data underlying the findings in their manuscript fully available?

Reviewer #1: Yes

Reviewer #2: Yes

Reviewer #3: Yes

4. Is the manuscript presented in an intelligible fashion and written in standard English?

Reviewer #1: Yes

Reviewer #2: Yes

Reviewer #3: Yes

5. Review Comments to the Author

Reviewer #1: This is a well performed study, with clear results and discussion. I would have a comment for the authors in the introduction - they state that: "despite advances in immunosuppression the ten-year survival rate remains stagnant due to lack of effective methods to target alloantibody". I do not completely agree with this statement because there is clear evidence from the CTS and other studies that long-term graft survival has significantly improved in comparison with graft survival in the 90 from the last century.

Reviewer #2: This study explores the effect of April and BlyS depletion in a rodent kidney transplant model. Using April -/- and BlyS-/- engineered rats, the authors described significant changes in antibody production, response to alloantigens, architecture of splenic germinal center and frequency of B cell and T cell subsets in various tissues, compared to WT animals. In sensitized transplanted April -/-/BlyS-/- rats, the authors found reduced alloantibody production, reduced early-stage and mature B lymphocytes, but not long-lived B lymphocytes, and less severe ACR and AMR, although the differences were not statistically significant. These results provide an insight into the effect of April and BlyS cytokines in a model of kidney transplantation and lay the ground for further studies exploring the value of these cytokines as therapeutic targets in transplantation.

I believe the following issues should be clarified:

1. Since depletion of early-stage and mature B lymphocytes has been observed in April -/-/BlyS-/- rats, is there any concern that B regulatory cell subsets may be among the depleted subsets? This has been a major concern in studies using B cell depleting agents, including studies using April/BlyS blockade (Bath et al, PLOS One, 2019).

2. The authors investigated the effect of April and BlyS depletion on alloantibody production by performing crossmatches with allogeneic T and B lymphocytes. Sera from non-sensitized and sensitized Lewis rats were used in these crossmatches. While testing sera from sensitized rats is meaningful, I am not clear why non-sensitized animals were also included. What level of alloantibodies was expected in these animals and what is the relevance of the crossmatch results?

3. Please provide representative flow cytometric data in the form of dot plots. That will help the reader to visualize and better understand the gating strategies, cell marker expression and the changes in cell subset frequencies.

4. The lack of data on transplant outcomes in animals who have not been previously sensitized should be listed as a limitation of the study.

Minor points:

1. Results section “April-/- and BlyS-/- lymphocyte proliferation significantly decreased when challenged with alloantigen” and Legend for Fig 4: For clarity, please specify that the responding lymphocytes were from Lewis rats.

2. The data presented in Figures 2 and 6 could be consolidated and presented in one figure.

3. The concluding remark of the Result section “Naïve and MZ B lymphocytes were significantly decreased in sensitized BlyS-/-“ is not clear. Please revise.

4. In the Result section “Splenic transitional zone (TZ) B lymphocytes significantly depleted by BlyS-/-“, the authors show that, in BlyS-/- animals, splenic TZ B cells were reduced while in other tissues these cells had similar frequency as observed in WT animals. Given these findings, it is not clear why the authors concluded that the absence of BlyS induced an accumulation of TZ B lymphocytes. Please clarify.

Reviewer #3: The article is well written and clearly demonstrates the findings based on the hypothesis and aims.

A few questions/comments

1. Please state what the abbreviation BCMA means.

2. How do you explain the difference in findings between the different lymphoid organs? eg lines 263-264?

3. The authors discuss their current findings and the use of belimumab in SLE. What study design would you employ to bridge the gap between the rodent model and clinical application? (Discuss next steps).

6. PLOS authors have the option to publish the peer review history of their article (what does this mean?). If published, this will include your full peer review and any attached files.

Reviewer #1: **Yes: **Assoc. prof. Antonij Slavcev, PhD

Reviewer #2: No

Reviewer #3: No

---

## [Author Response · Author response to Decision Letter 0]

25 Aug 2022

https://journals.plos.org/plosone/s/file?id=ba62/PLOSOne_formatting_sample_title_authors_affiliations.pdf".

These changes have been made.

These changes have been made to methods section.

“The authors would like to thank the UW CCC Flow Cytometry Shared instrumentation core, including the Shared Instrumentation grant 1S00OD018202-01 Special BD LSR Fortessa, which made possible the purchase and use of the BD LSR Fortessa.”

“RR- KL2 career development award (4KL2TR000428-10), https://ictr.wisc.edu/career-development-awards-2/; American Society of Transplant Surgeons Faculty Development Grant (MSN183242), https://asts.org/asts-foundation/grants-and-eligibility; American College of Surgeons Franklin Martin, MD, FACS Faculty Research Fellowship (MSN192116), https://www.facs.org/member-services/scholarships/research/acsfaculty.

Funding has been addressed in our cover letter.

Original gel has been uploaded as SF1.

Additional Editor Comments:

In each figure legend, please describe which test (ANOVA or T-test) was used.

This has been added to each figure as appropriate.

Reviewer #1: This is a well performed study, with clear results and discussion. I would have a comment for the authors in the introduction - they state that: "despite advances in immunosuppression the ten-year survival rate remains stagnant due to lack of effective methods to target alloantibody". I do not completely agree with this statement because there is clear evidence from the CTS and other studies that long-term graft survival has significantly improved in comparison with graft survival in the 90 from the last century.

This is a good point. Strides have certainly been made with regard to long-term allograft survival over the past 30 years. We have amended our statement to reflect that advances in graft survival have been made; however, there is still room for improvement (lines 48-50)

Reviewer #2: This study explores the effect of April and BlyS depletion in a rodent kidney transplant model. Using April -/- and BlyS-/- engineered rats, the authors described significant changes in antibody production, response to alloantigens, architecture of splenic germinal center and frequency of B cell and T cell subsets in various tissues, compared to WT animals. In sensitized transplanted April -/-/BlyS-/- rats, the authors found reduced alloantibody production, reduced early-stage and mature B lymphocytes, but not long-lived B lymphocytes, and less severe ACR and AMR, although the differences were not statistically significant. These results provide an insight into the effect of April and BlyS cytokines in a model of kidney transplantation and lay the ground for further studies exploring the value of these cytokines as therapeutic targets in transplantation.

I believe the following issues should be clarified:

1. Since depletion of early-stage and mature B lymphocytes has been observed in April -/-/BlyS-/- rats, is there any concern that B regulatory cell subsets may be among the depleted subsets? This has been a major concern in studies using B cell depleting agents, including studies using April/BlyS blockade (Bath et al, PLOS One, 2019).

This is an excellent point. We suspect that in the process of depleting all B cell subsets, B regulatory cells were likely also depleted as well. As B regulatory cells could help to decrease AMR, it is possible that by depleting these cells we are partially preventing immunological tolerance to the graft. This is one possible reason that no difference in AMR were seen between groups. This following statement has been included in the limitations (lines 602-606).

“Furthermore, it is possible that by targeting APRIL and BLyS, regulatory B lymphocytes, which play a role in inducing immunological allograft tolerance, are also depleted along with other B lymphocyte subsets. While we did not characterize regulatory B lymphocytes specifically, future studies should investigate changes in this subset as a potential explanation for no overall changes in AMR.”

2. The authors investigated the effect of April and BlyS depletion on alloantibody production by performing crossmatches with allogeneic T and B lymphocytes. Sera from non-sensitized and sensitized Lewis rats were used in these crossmatches. While testing sera from sensitized rats is meaningful, I am not clear why non-sensitized animals were also included. What level of alloantibodies was expected in these animals and what is the relevance of the crossmatch results?

As part of phenotyping these novel rodents, we wanted to (1) characterize B lymphocyte populations through various methods (immunospot, flow cytometry, histology) and to (2) assess what the functionality of these cells were, ie are they able to produce alloantibody. The purpose in performing these experiments including a crossmatch was to phenotype these animals at baseline prior to sensitization. While we did note a decrease in IgM and IgG secreting cells, this finding did not result in a decrease in alloantibody production. 

In normal wild type rodents, Brown Norway and Lewis rodents have a complete MHC mismatch meaning that regardless of sensitization status, Lewis rodents should produce alloantibody to Brown Norway. As shown in figure 3, non-sensitized BLyS-/- rodents produced less alloantibody in some DSA subsets, but this finding was inconsistent across DSA subclasses. Our hypothesis was that while DSA may not be significantly different between groups (and low to begin with), the changes in B cell subsets noted in non-sensitized animals could result in a change in DSA production even without sensitization. Our goal in performing a crossmatch in non-sensitized animals was to determine if there was any difference between WT, APRIL-/-, and BLyS-/- both in total MFI as baseline to compare to a sensitized model and to compare DSA production among 3 genetically different animals.

As expected in the sensitized setting, DSA production was much more robust and with this significant increase in DSA production, the significant difference in DSA production was able to be noted between WT, APRIL-/-, and BLyS-/-.

3. Please provide representative flow cytometric data in the form of dot plots. That will help the reader to visualize and better understand the gating strategies, cell marker expression and the changes in cell subset frequencies.

Flow cytometry dot plot demonstrating the preservation of TZ B lymphocytes has been included as Supplemental Figure 2.

Supplemental Figure 3 is a flow diagram of the gating for B cells since this is where gating is the most complex. If clarification or additional gating strategies are requested, please don’t hesitate to reach out.

4. The lack of data on transplant outcomes in animals who have not been previously sensitized should be listed as a limitation of the study.

This has been included at the end of limitations section (lines 618-620).

Minor points:

1. Results section “April-/- and BlyS-/- lymphocyte proliferation significantly decreased when challenged with alloantigen” and Legend for Fig 4: For clarity, please specify that the responding lymphocytes were from Lewis rats.

We have further clarified that WT, APRIL-/-, and BLyS-/- lymphocytes were from Lewis rodents (lines 303-304; lines 314-315).

2. The data presented in Figures 2 and 6 could be consolidated and presented in one figure.

We appreciate that fact that there are many figures in this paper (and often many individual graphs within one figure). We intentionally grouped data together examining these knock out animals in non-sensitized, sensitized, and finally transplant models. Since figure 2 represents non-sensitized animals and figure 6 marks the beginning of our sensitized animal data, we respectfully request that these two figures remain separate. Thank you for your consideration.

3. The concluding remark of the Result section “Naïve and MZ B lymphocytes were significantly decreased in sensitized BlyS-/-“ is not clear. Please revise.

Corrections have been made so that heading and caption now read “Sensitized BLyS-/- rodents demonstrated significantly fewer naïve and MZ B lymphocytes compared to sensitized WT and APRIL-/- rodents.”

4. In the Result section “Splenic transitional zone (TZ) B lymphocytes significantly depleted by BlyS-/-“, the authors show that, in BlyS-/- animals, splenic TZ B cells were reduced while in other tissues these cells had similar frequency as observed in WT animals. Given these findings, it is not clear why the authors concluded that the absence of BlyS induced an accumulation of TZ B lymphocytes. Please clarify.

The title for this sub-section is misleading. While splenic TZ B lymphocytes were decreased in BLyS-/-, we believe that the preservation of TZ B cells in other lymphoid tissues is a significant finding. As BLyS was not present for TZ B cells to differentiate, they appear to accumulate in lymph nodes, bone marrow and PBMC. The title for this sub-section has since been edited to reflect a preservation of lymph node and BM TZ B cells in BLyS-/-.

Reviewer #3: The article is well written and clearly demonstrates the findings based on the hypothesis and aims.

A few questions/comments

1. Please state what the abbreviation BCMA means.

Thank you for pointing out this omission. BCMA stands for B-cell maturation antigen. We have included this in text (line 62).

2. How do you explain the difference in findings between the different lymphoid organs? eg lines 263-264?

This is a great question and difficult to answer. In a conditional knock out model of APRIL/BLyS, it would make sense that mature B cells are less populous in secondary lymphoid organs (ie spleen and lymph nodes) since they have since migrated out of the bone marrow in order to become differentiated in response to antigen (ie lack of APRIL/BLyS prevents them from further differentiating in secondary organs) that have migrated to the spleen and lymph nodes from bone marrow. However, our model presented here is a germline knock out in which APRIL and BLyS were never present in these animals. Therefore, our hypothesis was that for BLyS and dual APRIL/BLyS knock out animals, all B cell lines would be depleted throughout all tissues. Additionally, since APRIL knock out animals still have BLyS present to help B cells to differentiate further, it is not surprising that B cells in APRIL knock out are not significantly changed. However, the reason for differences amongst spleen, bone marrow and lymph node will have to be further investigated.

3. The authors discuss their current findings and the use of belimumab in SLE. What study design would you employ to bridge the gap between the rodent model and clinical application? (Discuss next steps).

A pilot study investigating the safety and efficacy of anti-APRIL and anti-BLyS therapy in addition to the current standard of care regimen is likely the best next step to move therapy forward. The statement below has been added to the text.

“Clinical trials remain the gold standard to help progress treatment changes in transplantation. Data presented here and in the previously mentioned clinical trials provide further support to the existing data for the role of anti-APRIL and anti-BLyS therapy in transplantation (31). Dual therapy has shown promise in the pre-clinical setting and drug safety of current anti-APRIL and anti-BLyS medications have been established independent of each other. Therefore, next steps to move forward combination therapy may involve an open-label pilot study, in which the safety and efficacy of anti-APRIL/BLyS in addition to standard of care immunosuppression is investigated.

---

## [Decision Letter · Decision Letter 1]

21 Sep 2022

APRIL/BLyS Deficient Rats Prevent Donor Specific Antibody (DSA) Production and Cell Proliferation in Rodent Kidney Transplant Model

PONE-D-22-09163R1

Dear Dr. Bath,

We’re pleased to inform you that your manuscript has been judged scientifically suitable for publication and will be formally accepted for publication once it meets all outstanding technical requirements.

Kind regards,

Hodaka Fujii, M.D., Ph.D.

Academic Editor

PLOS ONE

Additional Editor Comments (optional):

Reviewers' comments:

Reviewer's Responses to Questions

**Comments to the Author**

1. If the authors have adequately addressed your comments raised in a previous round of review and you feel that this manuscript is now acceptable for publication, you may indicate that here to bypass the “Comments to the Author” section, enter your conflict of interest statement in the “Confidential to Editor” section, and submit your "Accept" recommendation.

Reviewer #1: All comments have been addressed

Reviewer #2: All comments have been addressed

Reviewer #3: All comments have been addressed

2. Is the manuscript technically sound, and do the data support the conclusions?

Reviewer #1: Yes

Reviewer #2: Yes

Reviewer #3: Yes

3. Has the statistical analysis been performed appropriately and rigorously? 

Reviewer #1: Yes

Reviewer #2: Yes

Reviewer #3: Yes

4. Have the authors made all data underlying the findings in their manuscript fully available?

Reviewer #1: Yes

Reviewer #2: Yes

Reviewer #3: Yes

5. Is the manuscript presented in an intelligible fashion and written in standard English?

Reviewer #1: Yes

Reviewer #2: Yes

Reviewer #3: Yes

6. Review Comments to the Author

Reviewer #1: No comments, the authors have adequately corrected the manuscript, I would suggest acceptance. I had only one remark concerning the long-term survival rate of organ transplants which has significantly improved in the last decades.

Reviewer #2: My comments were appropriately addressed by the authors. I believe that the revised version of the manuscript is acceptable for publication.

Reviewer #3: The authors have addressed the comments and questions from the reviewers.

The article in its current form is suitable for publication.

7. PLOS authors have the option to publish the peer review history of their article (what does this mean?). If published, this will include your full peer review and any attached files.

Reviewer #1: **Yes: **Antonij Slavcev, Assoc. prof., MD, PhD

Reviewer #2: **Yes: **Adriana Colovai

Reviewer #3: No

---

## [Editor Report · Acceptance letter]

3 Oct 2022

PONE-D-22-09163R1 

APRIL/BLyS deficient rats prevent donor specific antibody (DSA) production and cell proliferation in rodent kidney transplant model 

Dear Dr. Bath:

I'm pleased to inform you that your manuscript has been deemed suitable for publication in PLOS ONE. Congratulations! Your manuscript is now with our production department. 

Kind regards, 

on behalf of

Dr. Hodaka Fujii 

Academic Editor

PLOS ONE